# Vulnerability and agency in research participants' daily lives and the research encounter: A qualitative case study of participants taking part in scrub typhus research in northern Thailand

**Rachel C. Greer**[1,2]*, **Nipaphan Kanthawang**[1], **Jennifer Roest**[3,4], **Tri Wangrangsimakul**[1,2], **Michael Parker**[3,4], **Maureen Kelley**[3,4], **Phaik Yeong Cheah**[1,2,4]

**1** Mahidol Oxford Tropical Medicine Research Unit, Faculty of Tropical Medicine, Mahidol University, Bangkok, Thailand, **2** Centre for Tropical Medicine and Global Health, Nuffield Department of Medicine, University of Oxford, Oxford, United Kingdom, **3** Wellcome Centre for Ethics & Humanities, Nuffield Department of Population Health, University of Oxford, Oxford, United Kingdom, **4** The Ethox Centre, Nuffield Department of Population Health, University of Oxford, Oxford, United Kingdom

* rachel@tropmedres.ac

**Data Availability Statement:** All relevant data are contained within the paper. Complete interview

## Abstract

### Background

Researchers have a responsibility to protect all participants, especially vulnerable participants, from harm. Vulnerability is increasingly understood to be context specific, yet limited guidance is available regarding the vulnerability and agency of research participants in different cultural settings. This study aims to explore research participants' daily vulnerability and agency, and how these interact with participants' research experiences in their own words. Researchers' views and responses were also explored.

### Methods

A qualitative study was conducted around two scrub typhus research studies in northern Thailand. A thematic analysis was carried out on 42 semi-structured interviews with research participants, their families, researchers and key informants.

### Results

The majority of the research participants belonged to a hill tribe ethnic minority group. Common challenges were related to Thai language barriers, travel difficulties, uncertain legal status, unstable employment, lack of education and healthcare. We did not identify new vulnerabilities but we found that the extent of these vulnerabilities might be underestimated or even hidden from researchers in some cases. Despite these challenges people demonstrated agency in their daily lives and were often motivated and supported in this by family members. The majority of perceived research benefits were related to healthcare and gaining knowledge, while attending follow-up visits could be a burden for some.

data cannot be made publicly available due to ethical and legal reasons. These reasons relate to protecting the interests of research participants and data protection concerns. Due to the limited pool of participants and the inclusion of personal, sensitive information even the de-identified transcripts could compromise participant anonymity. The authors do not have blanket consent to share interview data openly without restrictions. The Mahidol Oxford Tropical Medicine (MORU) Data Access Committee as well as the Thai law on data protection have imposed these restrictions. Data access can be granted upon reasonable request from the MORU Data Access Committee, and a data access agreement will be put in place prior to data transfer. Instructions and the data application form are available here: https://www.tropmedres.ac/units/moru-bangkok/bioethics-engagement/data-sharing.

**Funding:** This study was supported by a Wellcome Trust and MRC Newton Fund Collaborative Award (200344/Z/15/Z), a Wellcome Trust Strategic Award (096527) and a Wellcome Trust Centre award (203132). The Mahidol Oxford Tropical Medicine Research Unit is funded by the Wellcome Trust (220211/Z/20/Z). The funders had no role in the study design, analysis or manuscript preparation.

**Competing interests:** The authors have declared that no competing interests exist.

**Abbreviations:** CIOMS, Council for International Organizations of Medical Sciences; GBP, British pounds sterling; LMIC, Low- and middle- income countries; RCT, Randomized-controlled trial; REACH, Resilience, Empowerment and Advocacy in Women's and Children's Health research project; UHC, Universal health coverage.

## Conclusions

Our approach to research in culturally and socioeconomically diverse settings should be more responsive to participants' specific vulnerabilities and abilities evidenced in their daily life, rather than attributing vulnerability on the basis of membership of pre-defined 'vulnerable groups'. Researchers need to be aware and responsive towards the challenges participants face locally in order to minimise the burdens of research participation whilst allowing participants to benefit from research.

## Introduction

The need to protect vulnerable research participants is well accepted in research ethics. However, little guidance is offered about how the experiences and impacts of vulnerability, and researchers' obligations might change in specific cross-cultural contexts. Research ethics guidelines have tended to focus on protecting participants' autonomy through the ensuring of informed consent [1]. Those who lack the capacity to consent, or cannot protect their own interests, have been labelled as vulnerable, alongside those who may be at risk of coercion, increased harm or wrong [2–5]. This paternalistic or stereotyping approach has led to many groups of people being identified as by definition vulnerable such as children, prisoners, those who are illiterate, and ethnic minorities [3–5]. Over time more vulnerable groups have been added, resulting in the concept losing much of its meaningfulness. While some groups may be protected, others could be unfairly excluded from research [5–8].

Recent critiques of group-based accounts of vulnerability have called for greater attention to the changing contexts in which research is being conducted, arguing that a group-based approach to vulnerability is too broad to meaningfully guide ethical decisions across the diverse cultural, political and socioeconomic contexts that we find in global health research [1, 5–7, 9–12]. Similarly, feminist accounts of vulnerability have argued for the importance of taking into account the abilities and agency of people and communities as well as the relational nature of agency through social networks [1, 9]. More nuanced conceptual accounts of vulnerability have been put forward, such as Luna's model of 'layers' of vulnerability which recognises that an individual can have more than one source of vulnerability [9, 10]. However, aside from giving 'special protection' [3] or 'special attention' [4] there is a lack of guidance on how ethics committees, researchers and funders should respond to research participants' vulnerabilities in context [1, 5, 8, 13].

There is a similar lack of empirical data relating to how agency should be identified and characterised throughout the research pathway. Historically, one's agency (the capacity to do, act, or take decisions) has at times been portrayed as the opposite of vulnerability and something that you either have or do not have [14, 15]. However, more recent accounts of agency introduce important nuances, suggesting that agency is something that can wax and wane [14, 15], is affected by the contexts people are in and is a continuum ranging from more hidden expressions such as resilience and coping, to more overtly bringing about change [14–17]. Agency can still be exercised in complex and challenging situations, though it should not be over emphasised as it may be constrained by circumstances [14–16].

In order to address some of this paucity in empirical data we conducted the Resilience, Empowerment and Advocacy in Women's and Children's Health research project (REACH). This collaborative research ethics study took place in Thailand, Kenya and South Africa to

investigate vulnerability and agency in different research contexts. In this paper, we present the data from Chiangrai, northern Thailand. Through a qualitative case study, we primarily aimed to characterise day-to-day research practice and ethical challenges in cultural context in a border region with large populations of hill tribe ethnic minority groups, many of whom might be considered vulnerable due to political status and socioeconomic circumstances. Secondly, we wanted to characterise how research participants' own account of daily challenges and struggles (vulnerabilities) and plans and decisions (agency), affected their experiences in research. Thirdly, we examined researchers' experiences of working with potentially vulnerable participants, their perceptions of vulnerability, and how they viewed their ethical obligations to respond in the research context. An in-depth analysis of the ethical considerations related to informed consent for research in northern Thailand will be reported separately.

## Study setting and methods

### The setting

Chiangrai is the northernmost province in Thailand and shares borders with Myanmar and Laos. In 2017, it had a population of 1.3 million people, the majority of whom live in rural areas [18]. Approximately 20% of the population are from a hill tribe ethnic minority group, the most common belonging to the Akha, Lahu and Hmong groups [19]. Hill tribe groups have different cultures, languages and traditions. They are commonly found living in the mountainous border regions of northern Thailand [20]. In the past, hill tribe people were considered to be non-Thai, despite many being second or third generation immigrants. It is only in recent decades that their Thai citizenship has been formally recognised. However, it is estimated that 12% to 50% still lack Thai citizenship [19, 21]. Thai citizenship is required for free movement, access to universal health coverage (UHC) and education, as well as rights to work, own land and vote. For those unable to prove their Thai citizenship, hill tribe identity cards were introduced in the 1990's which recognise their right to remain, but give less benefits than full citizenship, for example permission is needed to travel outside of the province [21, 22].

The average monthly income per household in Chiangrai Province is lower than the national average (375 GBP vs 648 GBP, respectively) and the average household debt in Chiangrai was 2,482 GBP in 2019 [23]. Over half of the working population are employed in agriculture [18], 12% of the population have no formal education and 50% have primary education or less [18].

Thailand has benefitted from UHC for its citizens since 2002. This has seen a reduction in some health inequalities, although more help is needed for those without cover and at risk populations, such as migrants and the poor [24–28].

Our study was based at the Chiangrai Clinical Research Unit (established in 2015) which is a small satellite unit of the Mahidol Oxford Tropical Medicine Research Unit (MORU). The main areas of research are scrub typhus, febrile illness, antibiotic use and bioethics. It is typically staffed by six to ten research staff; the majority of whom originate from Chiangrai province.

### Scrub typhus and the linked studies

The qualitative study reported here was based around two scrub typhus research studies (referred to as linked studies in the rest of the manuscript, see Table 1). Scrub typhus, a bacterial infection caused by *Orientia tsutsugamushi* is a leading cause of acute undifferentiated fever in the region [29–31]. Diagnosing scrub typhus can be challenging as most symptoms are non-specific (fever, cough, headache and malaise) and accurate point of care tests are lacking [32, 33]. If left untreated scrub typhus can be fatal. Transmission occurs through the bite of an

**Table 1. Linked scrub typhus study details.**

| Study details | | |
|---|---|---|
| Study title | The Scrub Typhus Antibiotic Resistance Trial comparing doxycycline and azithromycin treatment modalities in areas of reported antimicrobial resistance for scrub typhus | Eschar investigations to improve diagnostics, understand early immune responses and characterize strains for vaccines in scrub typhus |
| Study design | Randomized controlled trial (RCT) | Observational |
| Aims | Determine the optimum treatment for scrub typhus by comparing three oral antibiotic treatments | Improve understanding of the immune response to scrub typhus and investigate possible early diagnostics |
| Study population | Patients ≥ 15 years old hospitalised with non-severe scrub typhus | • ≥ 7 years old AND<br>• Patients presenting to hospital with scrub typhus OR<br>• Controls with skin injuries or attending minor surgery, who have had scrub typhus in the past or live in an endemic area. |
| Study processes | • Randomised to 1 of 3 treatment arms<br>• Demographic & clinical data<br>• Blood & urine samples at enrolment<br>• Daily clinical review while in hospital<br>• A further 6 or 12 blood samples over the next week<br>• Follow up at 2 and 8 weeks (clinical data, blood & urine samples) | Patients:<br>• Demographic & clinical data<br>• Eschar swabs, scrapings or biopsies<br>• Lymph node aspirates from a subgroup<br>• Blood & urine samples at enrolment<br>• Follow up at 2 weeks (clinical data, blood & urine samples)<br>Controls:<br>• Demographic & clinical data<br>• Blood & urine samples at enrolment<br>• Skin biopsies |
| Study benefits | • Treatment for scrub typhus (although most would be entitled to free treatment as part of routine care)<br>• Compensation for time and actual travel costs for enrolment and follow-up visits<br>• May help to improve scrub typhus treatment in the future | • No direct benefits<br>• Compensation for time and actual travel costs for enrolment and follow-up visits<br>• May increase understanding of scrub typhus disease severity and diagnostics |
| ClinicalTrials.gov identifier | NCT03083197 | NCT02915861 |

infected mite which is typically found in rural areas. Studies suggest that agricultural workers, those living in rural and mountainous areas, and belonging to a hill tribe ethnic group are disproportionately affected [34, 35].

## Methods

Our approach was an integrated ethics, case study design, wherein a small team of social science and bioethics researchers were linked with an ongoing clinical research programme in Chiangrai, Thailand [36–38]. The case was defined as scrub typhus research participants who had taken or were taking part in one of the linked studies (Table 1). It also included the perspectives of researchers involved in the studies and key community informants. Our study period ran from March 2018 to June 2019.

**Sampling and data collection.** Semi-structured interviews were conducted with a range of participants linked to the clinical studies. Participants were purposively selected from three groups: 1) female and child scrub typhus research participants and/or their family members, 2) researchers, ethics committee members and healthcare workers involved in the linked clinical studies, and 3) key community informants. Participants were identified from the linked research studies, existing networks and snowballing techniques.

We did not want to ascribe or impose the terms 'vulnerabilities' or 'agency' onto our participants' experiences but rather hear the descriptions in their own voices. Moreover, these words do not translate directly into Thai or the hill tribe languages; therefore we asked about participants' problems, difficulties and obstacles, as well as about their plans, decisions, and actions, drawing proxies for vulnerability and agency from the relevant literature on these topics [39].

Interview guides were created for Group 1 and, Groups 2 and 3 participants combined, with specific probes added for individuals, e.g. additional questions on the ethics review process were added for ethics committee members (see S1 and S2 Files). All groups were asked about their own (Group 1) or their perceptions of research participants' (Groups 2 and 3) challenges and sources of support in daily life and their research experience allowing for comparisons and triangulation to be made between the groups. In addition, Groups 2 and 3 participants were asked about their research experience and ethical situations that can occur during the research processes. The interview guides were created in English, translated to Thai with careful discussions of meaning with native speakers, and piloted for each group of participants. They were used as the basis of the interview with additional probes and questions asked as needed. Second interviews were conducted with some scrub typhus research participants, allowing for additional questions to be asked after reflections on the first interview.

The interviews were conducted in central and northern Thai dialects, or English depending on the participant's preferences by a native speaker. Both interviewers were female and based in Chiangrai. NK is an experienced Thai research nurse and RCG is a British research physician. Both are familiar with the context and have received training on qualitative research and interview skills. Interviews conducted in Thai (by NK) were translated into English allowing for additional questions to be asked (by RCG) if needed. Interviews were also conducted in hill tribe languages (Akha and Red Lahu) for those participants who were not fluent in Thai or English. In these interviews a trained and experienced translator conducted verbal translations between Thai and the relevant language.

Interviews were audio-recorded, transcribed verbatim and translated into English if needed. The immediate Thai translation for interviews conducted in hill tribe languages was transcribed and translated into English. Accuracy and clarifications were sought from the translators as needed.

**Analysis.** A thematic analysis was conducted using an iterative process taking a realist epistemological stance [40, 41]. Team members independently, inductively coded transcripts and discussed potential themes. We then developed a code book, drawing together inductive and deductive themes, the latter based on the study's aims and interview guides. These codes were subsequently applied to transcripts, discussed, and modified on an iterative basis over

**Table 2. Breakdown of REACH research participants.**

| Participants (number) | Interviews (number) | Other interviews (number) |
|---|---|---|
| **Group 1: Research participants & their family members from the linked scrub typhus studies (19)** | Research participants (14) Family members (5) | Second interviews (4: 3 with research participants, 1 with a family member) |
| **Group 2: Research staff from the linked scrub typhus studies (9)** | Research nurses (3) Senior research doctors (2) Hospital nurses (2) Ethics committee members and doctors (2) | Dyadic interview with 2 research nurses (1) |
| **Group 3: Key community informants (9)** | Primary care nurses (3, 1 is also a research nurse) Research nurse (1) Doctors and researchers (2) Village Chief (1) Director of a non-profit organisation (1) Informal translator & ex-village health volunteer (1) | |
| **Total = 37** | | |

multiple in-person and remote video call sessions. NK and RCG both coded all transcripts using NVivo Pro 11; any coding inconsistencies were discussed and clarifications were sought from the wider team. In addition, 10% of the transcripts were fully reviewed by another team member (PYC). The results are presented through a descriptive narrative approach. We intentionally focused on the research participants' experiences and triangulated these with the experiences and perceptions of researchers and community informants.

**Ethical approval and consent to participate.** This study was reviewed by the Chiangrai Public and Provincial Health Office Ethics Committee, Thailand (55/2560) and Oxford Tropical Network Ethics Committee, UK (OxTREC 534–17). Exemption was received from Chiangrai Hospital Ethics Committee, Thailand (CR.0032.102/research/17). All participants gave their written informed consent or parental consent and assent to participate. Assent and parental consent were obtained for those aged less than 18 years old. In Chiangrai, two witnesses are required for the consent process regardless of the participant's literacy level.

## Results

### Participant characteristics

In total, 42 semi-structured interviews were conducted with 37 participants (Table 2).

The demographic details of the REACH participants are shown in Table 3. The majority of the Group 1 participants were from a hill tribe group, some had shared Thai ethnicity, and most were Thai citizens. A fifth were born outside of Thailand. If you consider a group-based approach to vulnerability then the majority of these participants could be classed as being vulnerable, belonging to at least one of the following groups: children, ethnic minorities and low education.

A total of 6 people (4 research participants and their family members, and 2 ethics committee members) declined to join the study.

### Challenges and vulnerabilities in daily living

To understand the wider context of vulnerability and agency in research we began by characterising the challenges faced by research participants in their daily lives, how they cope with these challenges and how they exercised their agency. The way the research participants and their family members expressed their daily challenges varied; a minority talked explicitly about difficulties they faced. Others said they did not have challenges but on further questioning mentioned feeling anxious and worried about certain aspects of their lives, such as not having enough money. For others, seemingly challenging experiences were not described as such, being presented instead as simply part of normal life, shared by others in the same communities:

> '*We are [from a] hill tribe, so we usually work hard and have tough lives. . . It's normal in life.*' (*Scrub typhus RCT participant, 21–2*).

In these varied ways, participants communicated a range of challenging aspects to their lives, which interacted with and compounded each other. The most common ones were related to difficulties in communicating in the Thai language, and lack of legal status, education, employment and healthcare. Despite many services and support being available such as UHC and free education, peoples' ability to access them was not guaranteed, even for Thai citizens, as described below.

Language and effective communication are crucial for all areas of life, including access to services such as education and healthcare. Out of 19 participants, 13 were fluent in Thai, while

**Table 3. Demographic details of the REACH research participants.**

| Characteristic | Details | Number of participants, n (%) | | |
|---|---|---|---|---|
| | | **Group 1** | **Group 2** | **Group 3** |
| | | **Total = 19** | **Total = 9** | **Total = 9** |
| **Age** | < 18 years | 4 (21.1) | 0 | 0 |
| | 18–39 years | 6 (31.6) | 3 (33.3) | 3 (33.3) |
| | 40–54 years | 6 (31.6) | 4 (44.4) | 5 (55.6) |
| | ≥ 55 years | 3 (15.8) | 2 (22.2) | 1 (11.1) |
| **Sex** | Female | 16 (84.2) | 4 (44.4) | 7 (77.8) |
| | Male | 3 (15.8) | 5 (55.6) | 2 (22.2) |
| **Ethnicity** | Thai | 3 (15.8) | 7 (77.8) | 7 (77.8) |
| | Akha | 6 (31.6) | 0 | 0 |
| | Lahu (Black, Red, Yellow) | 7 (36.8) | 0 | 2 (22.2) |
| | Other hill tribe groups | 3 (15.8) | 0 | 0 |
| | White European | 0 | 2 (22.2) | 0 |
| **Fluent in Thai language** | Yes | 13 (68.4) | - | - |
| | No, need an interpreter | 6 (31.6) | - | - |
| **Education** | No formal education | 9 (47.4) | - | - |
| | Primary school | 5 (26.3) | - | - |
| | Secondary school | 2 (10.5) | - | - |
| | Higher education | 3 (15.8) | - | - |
| **Employment** | Agricultural work | 8 (42.1) | - | - |
| | Daily labour jobs | 3 (15.8) | - | - |
| | House maid | 2 (10.5) | - | - |
| | Vendors | 2 (10.5) | - | - |
| | Student | 2 (10.5) | - | - |
| | Retired | 1 (5.3) | - | - |
| | Unemployed | 1 (5.3) | - | - |
| **Legal status** | Thai or Thai citizen | 16 (84.2) | - | - |
| | Right to remain* | 2 (10.5) | - | - |
| | None | 1 (5.3) | - | - |
| **Health insurance** | Entitled to UHC | 16 (84.2) | - | - |
| | Some entitlement to UHC, have to pay for some investigations | 1 (5.3) | - | - |
| | No UHC | 2 (10.5) | - | - |

* Right to remain: legal status given to hill tribe members who are not Thai citizens but have a permanent right to remain in Thailand.

another six hill tribe participants needed a translator during our interview (Table 3). Those from a hill tribe, living remotely and with little education were less likely to be able to speak Thai and were reliant on their friends and families to translate for them:

> *'She can listen [to Thai language] but doesn't understand fully... when the doctor speaks there'll be words that are difficult to understand... [She] does not have the confidence in understanding, to understand the meaning [of it] like this. So she takes someone along with [her].' (Scrub typhus RCT participant, 14a, through an interpreter).*

Even with a translator, patients and healthcare workers were concerned about levels of understanding affecting the quality of care they were able to receive or give. Some Group 1

participants reported nodding along with the doctors and giving the impression that they understood when they did not.

*'[Being unable to speak Thai] is absolutely a problem because sometimes we may miss details of their health issues. We will prescribe only on the part that we understand from what they said.'* (Healthcare worker, 06).

Having some level of legal status e.g., Thai citizenship or the right to remain, is key to being able to work, attend school, access UHC and other governmental support. People without documents were fearful of travelling or meeting officials. Several of our hill tribe participants had received hill tribe identity cards, the right to remain, and were in the process of applying for Thai citizenship. Others had already received Thai citizenship and noted significant changes in their lives with their new citizen status. Only one of our participants had no legal status. However, participants from all three groups referred to other stateless, unregistered people who faced multiple challenges because of this and were unable to access state support.

*'She said before she was afraid, afraid to go anywhere. But now she has the [Thai identity] card, no more fear.'* (Scrub typhus RCT participant, 16, through an interpreter).

The majority of older Group 1 participants had not attended school compared with only one out of 10 participants under 40. In the past, the more remote sub-districts did not have schools, boys were given priority to study over girls, and children often had to stop studying to help provide an income for their family or look after younger children. A lack of or limited education affected many areas of people's lives, from day-to-day matters including employment, to the ability to vote and sign contracts. Healthcare workers felt that low education levels affected people's ability to understand their medical conditions and to look after themselves. Those from hill tribe backgrounds could particularly struggle with Thai language if they had not had the opportunity to study:

*'She said she didn't study so she doesn't know how things are. She doesn't understand anything because she stays only on the hills. She doesn't go any other places except the hills. She doesn't quite understand the language. . .has problems in communication and when working, she doesn't get a good job.'* (Mother of a scrub typhus observational study participant, 19b, talking about herself, through an interpreter).

Even now, despite schools being more accessible and education being provided free to citizens, the associated costs of travel, school equipment and financial demands limit some children's opportunities to study. Some mothers reported feeling bad for their children as they could not afford school uniforms and the financial burden of providing money for travel and education affected the whole family. These challenges affected Thai and hill tribe participants; raising enough money was a struggle, students had to work part time to support themselves and families had to live frugally. At times, people needed to sell financial assets, take out loans and rely on wider social support networks or foundations to provide for these financial needs. These challenges were corroborated by a village chief, who gave accounts of children having to stop studying so that they could help their parents work on the farms:

*'Children have problems, as children are in school only until 6th Grade and then have to stop to help parents with agricultural work, working in farms and fields. . . sending [them] to go to*

*study in higher levels. . . [they] don't have any [money] to send [them]. The parents struggle.'* (Village chief, 24).

Living remotely, often in mountainous areas meant service and infrastructure provisions were more limited, e.g., healthcare, schools, electricity. This usually meant that people needed to travel further to access some services which was particularly problematic in the rainy season and for those without their own vehicles. It also limited employment opportunities and meant some parents had to work away from home, leaving children and grandparents alone without any vehicles making their access to education and healthcare more challenging.

Group 1 participants typically worked in agriculture or had daily labour jobs, with associated job insecurities and low incomes. A couple of them explained how a shortage of money to invest reduced their productivity and ability to expand their work. A lack of crop diversity within households and communities meant that if a harvest failed or product prices were low that year then people would struggle financially.

Access to healthcare and people's treatment seeking behaviour were affected by many factors. Not being able to speak Thai and not having anyone to help translate was a barrier for many. Cost was raised as significant by both healthcare workers and Group 1 participants; those without Thai citizenship are not entitled to UHC and need to pay for their medical costs, this includes some hill tribe members living permanently in Thailand. Despite this, healthcare workers reported that everyone would be treated, costs would be minimised for those without coverage and support could be accessed from social welfare. Repaying hospital fees was possible through instalments and there was an acceptance that some fees would remain unpaid. However, availability of such support did not seem well known in the wider community and a few Group 1 participants were hesitant to seek care if they were not entitled to UHC.

Indirect medical costs such as travel were challenging for some, especially those who lived far away from healthcare and had no vehicle of their own. Group 1 participants' journeys to access healthcare could involve many steps and modes of transport such as walking, getting a lift, and taking public or private taxis. There was one example of someone who was entitled to UHC but did not receive it because they could not afford the cost of travel to the administration office to register their house move, so had to pay the hospital fees (approximately 50 GBP, UHC entitlements are linked to your place of residency).

Older Group 1 participants noted improvements over time in terms of healthcare access, and the quality and coverage offered by UHC. Inadvertently, this wider access meant that there could be long waits at the hospital resulting in healthcare being harder to access. Healthcare workers in particular felt that people overused the health system for minor illnesses. The time taken to seek care and the lack of out-of-hours provision presented a barrier to those unable to take time off work to seek medical care for themselves and their dependents.

These barriers to accessing healthcare meant that some Group 1 participants would delay seeking care, treat themselves or seek care at a lower level such as from traditional healers or clinics rather than the hospital. They reported delaying seeking care because of financial concerns, needing to take time off work, communication and transport challenges. Sometimes multiple challenges compounded each other causing further delays to treatment seeking and exacerbating health and other challenges. Two nurses commented, for example, that those without legal status sometimes delayed seeking care due to fears about crossing borders, the costs of treatment and language barriers.

*'There were cases where patients are afraid, afraid to come in. . . the legal issues, they crossed the border like this, they may be afraid that, if they come in, then they will be arrested, like*

*that. Therefore, they will come to us only when the [health] condition becomes severe. . . firstly the aspect of fear. . . Secondly, expenses. . .' (Healthcare worker, 27)*

Whilst strong social support was demonstrated and could reduce people's difficulties, dependency on others was also raised as a vulnerability, such as elderly patients relying on relatives to transport them or those from hill tribes needing others to translate for them. Middle aged Group 1 participants in particular spoke of the difficulties of balancing family responsibilities, such as caring for children and elderly relatives, as well as translating for and transporting relatives, looking after the house and earning money.

Healthcare workers, researchers, ethics committee members and other key informants identified a range of other factors that affected people's vulnerability, including being unable to make decisions for themselves or raise concerns about their challenges or difficulties. Some expressed concern about stigma towards hill tribe groups and diseases like human immunodeficiency virus. A key informant involved in an anti-trafficking not-for-profit organisation highlighted the danger of labelling people as being at risk:

*'I would try to avoid the term 'at risk' as much as possible when I interact with the kid. . . You don't have to name it. . . You don't have to say ah you're vulnerable. What's the point? . . .If there's any chance that it will make them feel less value or. . .less than others maybe it's better not to name it. . .So that they have their dignity, they have their self-value, and they don't have stigma.'* (Key community informant, 05)

### Coping and agency in daily living

Our findings demonstrate examples of agency in our participants' lives in a number of different ways. As with Payne's account of everyday agency [16], our Group 1 participants did not necessarily seem to see or describe their responses to experiences of hardship in terms of extraordinary actions, but instead presented these as part of normal life. They described making an effort not to over think difficulties. When asked how she coped with financial concerns one woman replied:

*'I don't think too much. Every day [I] live life frugally, grow vegetables and things to eat.'* (Scrub typhus RCT participant, 17, through an interpreter).

Despite experiencing challenging circumstances, we heard how people often took actions and made decisions to help other members of their families or communities. Family members travelled long distances, for instance, to care for relatives in hospital, despite not having their own vehicle and it being rainy season:

*'Well, if [you] ask [me] if it was difficult or not, well, it was difficult. But [it] was impossible for us not to go there, as there would've been no one there looking after her.'* (Aunt of a scrub typhus observational study participant, 18b).

Group 1 participants' decisions and actions were often motivated by their family roles and responsibilities and were made possible by the support of other family members. Parents strived to ensure that their children had the chance to study, especially if they were denied this themselves. Some parents travelled abroad, others left young children at home with their grandparents so that they could work, while a few had to sell their assets such as cows to

support their children's education. In this way people worked towards the enactment of their hopes and aims, albeit with limited options available to them; exercising agency but under constraint. Sometimes these choices had both positive and negative implications, and were judged poorly by others. One woman described how she worked overseas for several years while her children were young, so that they would be able to study in the future (an opportunity she herself did not have). This decision was not understood by her employees who felt she should have been home caring for her children, but she prioritised their education and was supported by her mother, so made these sacrifices:

> *'If my thoughts just revolved around warmth for [my] children, [my] children would never get to study. I went overseas [because] I had to work, so that my kids could study better, letting them face difficulties first [while I was away]. They were staying with their grandma... Well, I had to go anyhow, [if] I hadn't done that [I] wouldn't have been able to raise [my] children. [If] I had remained at home [it] would've been hard for [me] to find money, [making] me struggle, right, and my children wouldn't have been able to study to higher levels like this. So I thought, I'd rather leave to find a chunk of money for my children's studies like this [laughs].'*
> (Scrub typhus observational study participant, 10).

At the time of the interview, her oldest children had graduated from university and her youngest child was still studying with additional financial support from the ones who were working. Multiple participants described having to make similarly difficult decisions in the context of very limited opportunities. However, these constrained choices were presented positively, with expressions of optimism or acceptance. One girl was asked how she felt about having to stop studying early so that her younger family members could study and she said:

> *'Well, [I] don't regret anything. Yeah, at least, I know how to read and write.'* (Aunt of a scrub typhus observational study participant, 18b).

The main social support people relied upon was usually their families, with different members taking on varying roles. Owing to this dependency on family members, real challenges were faced by those without family support.

Other, broader social networks also provided strong, enabling support. There is a Thai tradition of '*aw mue*' or lending a hand, whereby people will help you with a task such as bringing in the harvest and you will reciprocate this at a later date. Villages had agreed fair, fixed rates for giving people lifts into the town, and payments could be made financially or through commodities like rice or working for the other person. It was also common place for people to borrow money from others until the next pay check or harvest. A few of the community key informants were actively working to help others in part due to the challenges that they had overcome, for example one orphaned man who had received an educational scholarship set up a foundation which helped other children access the education he had benefitted from himself. There were several examples of people from hill tribes becoming community leaders and working for the good of others.

Aspirations and hopes for the future were expressed, especially regarding education where it was hoped that not only the children's lives but also their parents' and future generations' lives would be improved:

> *'Even though she doesn't know about books [hasn't studied], she doesn't think about it because she sends her children to school already. If children graduate, grow up, they will be taking care*

*of the family. She hopes like that'.* (Scrub typhus RCT participant, 17 through an interpreter).

## How vulnerabilities and agency raised ethical considerations in research

The challenges people faced in their daily lives influenced their experiences of taking part in research and their perceptions of the benefits and burdens of research. Participation in research could affect family members as well as the participant, especially if they provided important sources of social support for the participant.

**Benefits of research.** All but one Group 1 participant reported benefits from taking part in research. The other RCT participant reported no benefits and said participating in research felt 'normal'. The majority were related to their own healthcare which may reflect people's limited or challenging access to healthcare. The following benefits were expressed by participants from both the observational study and RCT. Most felt they gained knowledge about what was wrong with them, how they needed to be treated and how to prevent scrub typhus. Participants appreciated the tests and diagnoses given to them as part of research:

> *'[I] was happy when I went for the test and [found that] I had nothing. . . I found out what disease I had, what disease I didn't have as they knew all [about it] having drawn blood and so on.'* (Scrub typhus RCT participant, 03).

Half of the Group 1 participants said that a benefit of research was being cured or treated, although the vast majority were entitled to UHC regardless of research participation. They felt they were well taken care of, received extra attention and follow-up visits in addition to standard care because they took part in the research.

Several Group 1 participants saw the compensation as a benefit. They mentioned that being compensated financially or receiving travelling costs was helpful. They also appreciated research staff travelling to meet them:

> *'It helps her a lot in many ways. . . providing budget for travelling. . .it helped my son get well. That was a lot of help. . . the research team came to help so she didn't have to travel far downtown. . . And they came here, home [visit] service, like oh! That was already great.'* (Mother of a scrub typhus observational study participant, 19b, through an interpreter).

In addition to individual benefits, around a third of Group 1 participants thought that research would benefit others in the future, by improving knowledge about treatments. A scrub RCT participant thought it may help to prevent the disease and raise awareness amongst people who live in the mountains:

> *'They will be able to study from her. Nurses will get this knowledge to improve later. . . to treat others in the future.'* (Scrub typhus RCT participant, 17 through an interpreter).

These research benefits were similarly expressed during the Groups 2 and 3 interviews; they felt that the linked study participants' health could be improved, that knowledge would be gained and research could benefit others in the future.

**Burdens of research.** In this cultural context, where graciousness and politeness is paramount, it was difficult to get people to talk about something being a burden. The majority of Group 1 participants initially said that joining the research did not cause them any burdens.

On further questioning, however, the most commonly reported burdens were related to attending the follow-up visits. Travel was difficult if people did not have access to their own vehicle or needed to rely on others to take them. Some needed family members to accompany them due to travelling concerns or language barriers. This could result in higher costs, family members missing work and burdens on family members' as well as the research participant's time.

Most researchers, particularly frontline staff were aware of the challenges associated with follow-up visits and tried to limit their impact by meeting the participants after hospital appointments, or close to home, and at convenient times which meant some did not need to travel far, or miss work and school. Participants seemed to appreciate this and chose to meet the researchers part way:

> *'She thought we [research team] might not be able to come see them [19b's family] because the roads may be difficult for us. We might not be used to the directions so they chose to travel to meet us instead like this.'* (Mother of a scrub typhus observational study participant, 19b, through an interpreter).

The compensation given to research participants seemed to confuse and concern some; they felt they must be being paid for something or that something bad was going to happen. Some noted that it was not normal to receive money so worried why they were being given money:

> *'Coming to do the research study, coming to give the treatment, why did [they] have to bring money for [her], did [they] come [here] to buy something? At times, she said she was worried and pessimistic, thinking, err, if [they] were going to sell [or] do something or not, like this.'* (Sister-in-law of a scrub typhus RCT participant, 14 b via an interpreter).

Other burdens mentioned by several participants included potential medication side effects and having blood tests. Overall, participants from all groups seemed to feel that the benefits of research outweighed the burdens:

> *'She thinks blood draws cause weakness. But she doesn't worry much. It's ok because they must get the blood test for medical treatment.'* (Scrub typhus RCT participant, 17, through an interpreter).

Participants from the observational study and RCT described similar burdens. However, confusion and concerns surrounding the compensation given were expressed solely by two participants from the RCT.

## Discussion

Our study revealed valuable insights into the context of day-to-day research with participants who, on the face of it, would be considered vulnerable on multiple counts. We did not identify new vulnerabilities. However, from our findings, it appears that the extent of these vulnerabilities might be underestimated or even hidden from researchers in some cases. We also found that despite these vulnerabilities, linked study participants benefitted from participating in research such as gaining knowledge about their own health and benefiting others in the future. They also demonstrated agency and made choices, hoping for a better future. Research participants drawn from scrub typhus patients in northern Thailand can face many challenges in their daily lives, related to language barriers, legal status, employment, and accessing education

and healthcare. These specific challenges can exacerbate and reinforce each other, worsening overall vulnerabilities.

The research participants' daily challenges were reflected in and appeared to influence their experiences of research. Some of these dynamics were apparent, and research studies in the area tend to already be cognisant of them, such as challenges related to living far away from the hospital or difficulty speaking Thai. With regards to travel, for instance, in addition to long distances, many Group 1 participants did not own a vehicle, and so either had to pay high costs to hire one or had many steps in their journey to the hospital. Difficulties owing to language barriers were also not necessarily clear, as some participants gave the impression that they followed the conversation in Thai, when they were only able to understand the main points. In response to this challenge a formal network of translators was established and trained during our data collection phase. At the time of writing, the linked clinical studies are continuing to utilise these translators for participants unable to speak Thai fluently, rather than relying on informal translators (e.g. friends or family members) as is standard practice locally.

Despite the difficulties people faced, many Group 1 participants considered their challenges normal and no different to others. They demonstrated agency and made choices, hoping for a better future for themselves and their family, similar to the 'everyday agency' Payne describes from her work in Zambia with child head of households. The children viewed their new roles and responsibilities for younger siblings as normal rather than seeing themselves doing something extraordinary [16]. As in other settings, people's agency was often constrained by their situations and their choices at times were limited [14–17]. Individuals often prioritised the interests of their children and other family members in decision-making, such as the mother who chose to work overseas and be separated from her children so that her children could study [14]. People's agency appeared to be enabled by those around them, particularly the support they could draw on from their families and the wider community. People were better able to access health services and take part in research because of the assistance they received from family members, especially with regards to interpreting and understanding the proceedings. In these ways, our findings demonstrate multiple expressions and sources of agency in people's lives, despite experiences of vulnerability. They also point to the importance of recognising and supporting agency through the research process, being especially responsive to a person's particular capacities. Most people in our study felt fully able to make decisions for themselves, though could potentially be better supported–particularly around language–to do so.

The linked scrub typhus studies are examples of a locally important research topic which can disproportionally affect the rural poor, and in this setting, hill tribe groups, and may not be feasible to conduct in other populations [34, 35]. Many of the social determinants of health stem from being disadvantaged, such as poverty and lack of education [1, 42]. The optimal antibiotic treatment for scrub typhus in this region is unclear and challenges in its diagnosis can lead to delays in treatment and more severe illness [32, 33, 35]. Without research these gaps in knowledge will remain.

Similar to other studies from low- and middle- income countries (LMICs), receiving treatment [39, 43, 44] and knowledge were the main benefits for participants taking part in this research, and some hoped their research participation would benefit others in the future. It is important that such benefits are not denied to groups of people who may be labelled as being vulnerable by fully excluding them from research [3, 5, 45, 46]. Their agency and ability to make decisions, including to participate in research, should be respected, even if it is constrained by their circumstances [1, 11].

## Implications

Our data support the growing call to rethink the concept of research vulnerability away from group-based classifications to more nuanced understandings and approaches that take into account participants' situations and abilities [1, 5, 6, 9–13, 39]. Researchers and ethics committees need to consider the daily challenges that participants face and be aware that circumstances can change, and not all will face the same challenges. This understanding of the specific, individual challenges faced in local contexts is required in order to design studies and minimise the potential harms and burdens research participants may face [10]. Researchers need to be particularly aware of hidden vulnerabilities and respond to any that may arise during the study period; this is likely to require additional ethics support. This awareness will help to reduce the risks of creating or aggravating vulnerabilities [39]. Our results demonstrate how researchers can minimise the burdens of follow-up, by often meeting participants half way or scheduling appointments to match the hospital's schedule. Meeting actual travel costs and giving compensation for participants' time is important despite concerns about undue inducements. Studies need to be adequately funded and staffed in order to reduce the burdens of follow-up and travel usually placed on the participants. Addressing participants' vulnerabilities will also help to respect their agency and support them to make decisions about their research participation.

The results of this study can be used to provide training for researchers working in similar areas to raise awareness of the challenges research participants can face in daily life and while taking part in research. During the recruitment process researchers can ask participants about any challenges they foresee about participating in the study and work together to find possible solutions. Our results combined with ongoing engagement with the community and frontline research staff will help to design studies with further context specific adaptations to minimise research burdens while maximising benefits for participants. Regular review and reflection sessions during study implementation may help to identify unexpected burdens or hidden vulnerabilities and provide a forum to discuss possible actions and solutions [47]. While we did not set out to draw clinical or public health implications from this work many of the findings are relevant to clinicians working in the area, particularly the barriers to accessing healthcare such as the need for formal translators, indirect medical costs and the costs of care for those not entitled to UHC.

## Strengths and limitations

Our research adds to the limited empirical data related to vulnerabilities and agency of research participants in global health research. The case study design allowed us to gather rich contextual data and experiences from a range of participants related to ongoing clinical research. Understanding the different perceptions of research participants, researchers, ethics committee members and community leaders is important to address this complex challenge holistically. The influence of context on one's vulnerability and agency and our focus on women and children limits the transferability of some of our findings, however people working in similar situations will be able to identify with many of our findings such as the challenges of language barriers. This study has some important limitations; due to the slow recruitment and wide catchment area for the provincial hospital it was not possible to form focus group discussions of research participants, so the data is based on individual interviews. The majority of research participants were entitled to UHC so we may have underestimated the burdens of accessing healthcare for those who are excluded from the coverage, such as temporary migrants and unregistered people.

## Conclusions

Research driven by local priorities and health needs is important, especially in LMICs. In order to carry out such research ethically researchers, ethics committees and funders need to be aware of the local context and specific vulnerabilities that research participants may face. However, the fact that communities and the individuals within them have vulnerabilities does not in itself mean that research should not be undertaken with them. Research needs to be designed to limit the burdens faced by participants and to be responsive to hidden or unexpected vulnerabilities that arise during study participation, this could be achieved through effective community engagement and reflective discussions within research teams. Efforts also need to be made to respect, and where possible, enhance the scope for participants and potential participants to express agency in relation to decisions about research participation. Key to this is that individual vulnerabilities and abilities need be considered rather than approaches which attribute vulnerability on the basis of membership of pre-defined 'vulnerable groups'.

## Supporting information

**S1 File.**
(DOCX)

**S2 File.**
(DOCX)

## Acknowledgments

We wish to thank the participants and collaborating scientists for their time and helpful insights, as well as Napat Khirikoekkong, Suphak Nosten, Supa-at Asarath, Supalert Nedsuwan and Daranee Intralawan for their support and guidance on this project. We would like to thank Khachornphit Wongyai and Janchao Prukpongsawalee for translating the Thai interview transcripts into English, and the Akha and Red Lahu interpreters.

## Author Contributions

**Conceptualization:** Michael Parker, Maureen Kelley, Phaik Yeong Cheah.

**Data curation:** Rachel C. Greer, Nipaphan Kanthawang.

**Formal analysis:** Rachel C. Greer, Nipaphan Kanthawang, Jennifer Roest, Maureen Kelley, Phaik Yeong Cheah.

**Funding acquisition:** Michael Parker, Maureen Kelley, Phaik Yeong Cheah.

**Investigation:** Rachel C. Greer, Nipaphan Kanthawang, Maureen Kelley.

**Methodology:** Rachel C. Greer, Jennifer Roest, Maureen Kelley, Phaik Yeong Cheah.

**Project administration:** Tri Wangrangsimakul, Maureen Kelley, Phaik Yeong Cheah.

**Resources:** Tri Wangrangsimakul, Phaik Yeong Cheah.

**Supervision:** Phaik Yeong Cheah.

**Validation:** Rachel C. Greer.

**Visualization:** Rachel C. Greer.

**Writing – original draft:** Rachel C. Greer.

**Writing – review & editing:** Rachel C. Greer, Nipaphan Kanthawang, Jennifer Roest, Tri Wangrangsimakul, Michael Parker, Maureen Kelley, Phaik Yeong Cheah.

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
