## [Decision Letter · Decision Letter 0]

25 May 2022

PONE-D-21-20925Vulnerability and agency in research participants’ daily lives and the research encounter: A qualitative case study of participants taking part in scrub typhus research in northern ThailandPLOS ONE

Dear Dr. Cheah,

Thank you for submitting your manuscript to PLOS ONE. After careful consideration, we feel that it has merit but does not fully meet PLOS ONE’s publication criteria as it currently stands. Therefore, we invite you to submit a revised version of the manuscript that addresses the points raised during the review process.

Please note that we have only been able to secure a single reviewer to assess your manuscript. We are issuing a decision on your manuscript at this point to prevent further delays in the evaluation of your manuscript. Please be aware that the editor who handles your revised manuscript might find it necessary to invite additional reviewers to assess this work once the revised manuscript is submitted. However, we will aim to proceed on the basis of this single review if possible.  Your manuscript has been assessed by an expert reviewer, whose comments are appended to this letter. As you will see, the reviewer has raised some concerns about the reporting of the methodology and the balance of perspectives from the different groups. Please ensure you address these points, and the others detailed in the report below, in your response to reviewers document, and revise your manuscript accordingly.

We look forward to receiving your revised manuscript.

Kind regards,

Joseph Donlan

Editorial Office

PLOS ONE

Journal Requirements:

2. When reporting the results of qualitative research, we suggest consulting the COREQ guidelines: http://intqhc.oxfordjournals.org/content/19/6/349. In this case, please consider including more information on the number of interviewers, their training and characteristics; and please provide the interview guide used.

Reviewers' comments:

Reviewer's Responses to Questions

**Comments to the Author**

1. Is the manuscript technically sound, and do the data support the conclusions?

Reviewer #1: Yes

2. Has the statistical analysis been performed appropriately and rigorously? 

Reviewer #1: N/A

3. Have the authors made all data underlying the findings in their manuscript fully available?

Reviewer #1: Yes

4. Is the manuscript presented in an intelligible fashion and written in standard English?

Reviewer #1: Yes

5. Review Comments to the Author

Reviewer #1: This manuscript presents qualitative data about daily life challenges and how agency is exercised among a hill tribe ethnic minority in Thailand, particular as they relate to participation in biomedical research relevant to this group (i.e. scrub typhus studies). The manuscript is well-organized, clearly written and should be of some interest to researchers, research ethics committee members and ethicists interested in issues related to the inclusion of ‘vulnerable’ populations in research in low-and-middle income countries (LMICs). Major and minor comments are detailed below.

Major comments

A possible criticism of the manuscript is that it does not seem to provide much new information to the ongoing discussion about biomedical research in LMICs with marginalized communities. It has long been known that social determinants adversely impact health as well as willingness/ability to participate in research that might benefit such communities. On the other hand, it is likely that such a descriptive account has not been done before with this particular community, and certain barriers (particularly linguistic, citizenship status and access to health care benefits) may resonate with experiences of other communities in LMICs.

The methods section (lines 126-134) does not really convey the methods. Only by line 180 is it clear that semi-structured individual interviews were conducted. There are many ways of doing case studies, so what did ‘case study design’ involve here? There is little sense of what kinds of questions were asked to the three different groups, including whether they were asked the same sorts of questions in order to make comparisons.

It would seem that although two different research studies were involved, the results (in the form of illustrative quotes or descriptions) are lumped together. This makes it hard to know when (for example) people talk about future benefits of research or personal benefits, they are talking about the observational study or the randomized controlled trial.

The participants make up the bulk of the respondents. In the reporting of the results, they also seem to be lumped together, i.e. it is unclear which of the participants participated in the observational study or the randomized controlled trial, or both. This way of reporting may conceal differences in experiences of participating in these studies.

Although three groups were interviewed, the perspectives of research participants and family members are disproportionately respresented. Is this intentional? I counted 16 illustrative quotes from research participants and their family members, while there was only 1 from group 2, and 1 from group 3. The views of those in groups 2 and 3 seem to be reported in aggregate, with little indication of tensions or differences of opinion between group members or with the views of participants (perhaps there was a lot of consensus?).

Was the formal network of translators established as a result of the qualitative study reported in this manuscript or was it already established prior to that? It is hard to tell from lines 568-571.

In the conclusion section, where it says research needs be designed to limit the burdens and respond to unexpected vulnerabilities (lines 641-643), it may be advisable to say something about approaches to do that, particularly in the context they describe. Would it include social science research in advance of the implementation of biomedical trials? Some form of community engagement? Something else?

Minor comments

What is meant by ‘linked’ when the two studies are described as linked?

Line 542: “Overall, people …” Does this mean all interviewees across the three groups interviewed? This comes in after describing research participant views, so please clarify.

Line 80: should probably be “while some may be protected, others could be unfairly …”

Lines 352-354: there seems to be either extra words or words missing in this sentence. Please recheck.

6. PLOS authors have the option to publish the peer review history of their article (what does this mean?). If published, this will include your full peer review and any attached files.

Reviewer #1: No

---

## [Author Response · Author response to Decision Letter 0]

12 Sep 2022

see Response to Reviewers document

---

## [Decision Letter · Decision Letter 1]

27 Oct 2022

PONE-D-21-20925R1Vulnerability and agency in research participants’ daily lives and the research encounter: A qualitative case study of participants taking part in scrub typhus research in northern ThailandPLOS ONE

Dear Dr. Greer,

Thank you for submitting your manuscript to PLOS ONE. After careful consideration, we feel that it has merit but does not fully meet PLOS ONE’s publication criteria as it currently stands. Therefore, we invite you to submit a revised version of the manuscript that addresses the points raised during the review process. Please submit your revised manuscript by Dec 11 2022 11:59PM. If you will need more time than this to complete your revisions, please reply to this message or contact the journal office at plosone@plos.org. Please include the following items when submitting your revised manuscript:A rebuttal letter that responds to each point raised by the academic editor and reviewer(s). You should upload this letter as a separate file labeled 'Response to Reviewers'.A marked-up copy of your manuscript that highlights changes made to the original version. You should upload this as a separate file labeled 'Revised Manuscript with Track Changes'.An unmarked version of your revised paper without tracked changes. You should upload this as a separate file labeled 'Manuscript'.

We look forward to receiving your revised manuscript.

Kind regards,

Ahmed Mancy Mosa, Ph.D.

Academic Editor

PLOS ONE

Additional Editor Comments:

Please consider all comments

Reviewers' comments:

Reviewer's Responses to Questions

**Comments to the Author**

1. If the authors have adequately addressed your comments raised in a previous round of review and you feel that this manuscript is now acceptable for publication, you may indicate that here to bypass the “Comments to the Author” section, enter your conflict of interest statement in the “Confidential to Editor” section, and submit your "Accept" recommendation.

Reviewer #2: All comments have been addressed

Reviewer #3: (No Response)

2. Is the manuscript technically sound, and do the data support the conclusions?

Reviewer #2: Yes

Reviewer #3: Partly

3. Has the statistical analysis been performed appropriately and rigorously? 

Reviewer #2: Yes

Reviewer #3: N/A

4. Have the authors made all data underlying the findings in their manuscript fully available?

Reviewer #2: No

Reviewer #3: Yes

5. Is the manuscript presented in an intelligible fashion and written in standard English?

Reviewer #2: Yes

Reviewer #3: Yes

6. Review Comments to the Author

Reviewer #2: Manuscript #: PONE-D-21-20925R1

Topic: Vulnerability and agency in research participants’ daily lives and the research encounter: A qualitative case study of participants taking part in scrub typhus research in northern Thailand.

Academic Editor: Dr. Ahmed Mancy Mosa

Thanks very much, Dr. Ahmed Mancy Mosa, for giving me the opportunity to review and give comments. My thanks also go to the authors of the manuscript for their interest to deal with this interesting topic addressing the issue of vulnerability and agency in research participants which are the areas that need more research to come up with concrete and convincing shreds of evidence. I am impressed with the title, and the way the authors synthesize and present the result. The whole section of the manuscript has been well prepared and comes up with interesting findings. In fact, this is the second version of the corrected manuscript based on the first comments given by the other reviewers, the authors come up with a well-organized and appropriately synthesized document. I have only one question as well as a suggestion to be addressed by the authors before the manuscript will be published. What are the clinical and public health implications of this research? Please, incorporate the clinical, and public health implications of the findings of this research.

Reviewer #3: Thanks to the authors for showing us the daily vulnerabilities and initiatives of participants in scrub typhus research in northern Thailand and how they interact with participants' own research experiences. Overall, this manuscript is interesting. But there are plenty of concerns.

1. The implications of the study are unclear. The extensibility and extensibility of the research are not enough.

2. The definition of some words in the manuscript is not clear, which will cause unnecessary trouble to readers. For example, what does “agency” mean in the title and START study and EXIST study in Table 1?

3. Key words: The keyword global health research is not reflected in the manuscript.

4. Introduction: There's so much in this section, it's more like a review. It is recommended to simplify. Study setting and methods:

a) A large number of references to scrub typhus and related studies are not recommended for inclusion in this manuscript.

b) Tables 2 and 3 are recommended for the results section. The obvious Table 3 describes the demographic characteristics of the first group of subjects. Why does it not reveal the most basic demographic trait, sex? Why not describe the demographic characteristics of the subjects in the second and third groups? The grouping of Education, Employment and Legal status in Table 3 is not clear. For example, Primary school or less and None in Education may coincide, studying refers to students in Employment? What does "Right to remain" mean in Legal status?

5. Results:

a) This study adopts thematic analysis, but it is difficult to see the inductive theme in the results.

b) Lines 253-258 seem to describe the purpose of the study and are suggested to be put in the background.

c) The responses of the other two groups were used to support the analysis of the first group. For example, in line 325-339, the first group is analyzed but the answers of the third group are used as evidence.

d) Lines 385 to 387, using the two nurses as an example, are not supported by evidence.

e) The contents of Benefits of research and Burden of research do not seem to have much relevance to this study, but more to describe the benefits and burdens of the previous two studies on scrub typhus.

6. Discussion:

a) The authors suggest redefining vulnerability. However, there are no more comments and suggestions in the discussion section.

b) The examples on lines 588-591 do not fit in the discussion section.

7. References: References that are too old are of little value, especially 6-9 and 11.

7. PLOS authors have the option to publish the peer review history of their article (what does this mean?). If published, this will include your full peer review and any attached files.

Reviewer #2: **Yes: **Gossa Fetene Abebe

Reviewer #3: No

---

## [Author Response · Author response to Decision Letter 1]

8 Dec 2022

Please see attached file for our response to the reviewers

---

## [Decision Letter · Decision Letter 2]

20 Dec 2022

Vulnerability and agency in research participants’ daily lives and the research encounter: A qualitative case study of participants taking part in scrub typhus research in northern Thailand

PONE-D-21-20925R2

Dear Dr. Greer,

We’re pleased to inform you that your manuscript has been judged scientifically suitable for publication and will be formally accepted for publication once it meets all outstanding technical requirements.

Kind regards,

Ahmed Mancy Mosa, Ph.D.

Academic Editor

PLOS ONE

Additional Editor Comments (optional):

Reviewers' comments:

Reviewer's Responses to Questions

**Comments to the Author**

1. If the authors have adequately addressed your comments raised in a previous round of review and you feel that this manuscript is now acceptable for publication, you may indicate that here to bypass the “Comments to the Author” section, enter your conflict of interest statement in the “Confidential to Editor” section, and submit your "Accept" recommendation.

Reviewer #2: All comments have been addressed

Reviewer #3: All comments have been addressed

2. Is the manuscript technically sound, and do the data support the conclusions?

Reviewer #2: Yes

Reviewer #3: Yes

3. Has the statistical analysis been performed appropriately and rigorously? 

Reviewer #2: Yes

Reviewer #3: Yes

4. Have the authors made all data underlying the findings in their manuscript fully available?

Reviewer #2: Yes

Reviewer #3: Yes

5. Is the manuscript presented in an intelligible fashion and written in standard English?

Reviewer #2: Yes

Reviewer #3: Yes

6. Review Comments to the Author

Reviewer #2: (No Response)

Reviewer #3: Thanks to the authors for showing us the revised manuscript. The author addressed the problems raised by the reviewer. We find the manuscript acceptable.

7. PLOS authors have the option to publish the peer review history of their article (what does this mean?). If published, this will include your full peer review and any attached files.

Reviewer #2: **Yes: **Gossa Fetene Abebe

Reviewer #3: No

---

## [Editor Report · Acceptance letter]

4 Jan 2023

PONE-D-21-20925R2 

Vulnerability and agency in research participants’ daily lives and the research encounter: A qualitative case study of participants taking part in scrub typhus research in northern Thailand 

Dear Dr. Greer:

I'm pleased to inform you that your manuscript has been deemed suitable for publication in PLOS ONE. Congratulations! Your manuscript is now with our production department. 

Kind regards, 

on behalf of

Dr. Ahmed Mancy Mosa 

Academic Editor

PLOS ONE